


# RICHARD 1.0 - Routine for the Isolation of Chemical Hotspots in Atmospheric Research Data

Christian Scharun[1], Roland Ruhnke[1], and Peter Braesicke[1]

[1]Karlsruhe Institute of Technology, Institute of Meteorology and Climate Research - Atmospheric Trace Gases and Remote Sensing, Karlsruhe, Germany

**Correspondence:** Christian Scharun (christian.scharun@kit.edu)

**Abstract.** Here, we introduce version 1.0 of the RICHARD algorithm, a **R**outine for the **I**solation of **C**hemical **H**otspots in **A**tmospheric **R**esearch **D**ata, e.g. in satellite measurement datasets (level-2 and above) or atmospheric chemistry model output. The overall goal of the algorithm is to identify "hotspot" areas in which local enhancements of an atmospheric constituent can only occur due to strong local emissions. To detect hotspot areas, we use a mathematical method that combines spatiotemporal

proxy data for calibration purposes and a selection algorithm in a novel way. For each input file family (e.g. the near surface mixing ratio or the tropospheric partial column of atmospheric constituents as a function of longitude and latitude) we define a structure quotient by which the algorithm decides - based on a threshold value - whether at a particular iteration step the hotspot area criteria are met and if they are kept for further calculations or not. The python based command line tool RICHARD 1.0 comes with a set of implemented features like an automated generator for user-defined patterns and an analysis tool to determine

the optimal threshold value for a given dataset.

For testing purposes of RICHARD 1.0, we use simulations of the atmosphere and chemistry modeling framework ICON-ART, a joint development of the German Weather Service and Max-Planck-Institute for Meteorology in Hamburg. We comprehensively explore different aspects of RICHARD using ICON-ART model output datasets. We provide an analysis of the decision making process coded in RICHARD, and provide a detailed look at the competing effects of emissions and advection.

Here, we also consider the direction and speed of the wind that affect the advection of prescribed (and thus known) emissions in the model and look at the resulting tracer mixing ratios as to evaluate the sensitivity of the algorithm and its ability to identify objectively hotspots of strong emissions, based on the self-determined threshold values.

The results show that RICHARD can identify frequently (or continuously) emitting localised sources as hotspots. Furthermore, the algorithm is able to distinguish between an actual emission source and other circumstances that lead to enhanced

tracer concentrations, e.g. as caused by wind conditions and associated transport processes. In addition, the in ICON-ART prescribed emission source strengths are detected and quantified regardless of overlying transport features with only a small error of about 5 %. This increases significantly the accuracy of determined source strengths compared to other methods that we have explored.

RICHARD 1.0 is a novel comprehensive tool for the identification and quantification of emission hotspots and uses a novel

workflow that includes spatiotemporal proxy data as well as a selection algorithm. Here, we present a model-based proof of concept that is already fully transferable to applications using satellite measurement data.



# 1 Introduction

The detection, attribution and quantification of greenhouse gas (GHG) emissions is an important task for monitoring mitigation strategies under climate change. How to achieve robust estimates of GHG emissions is frequently discussed in many sub-disciplines of climate research like modelling, the observation and measurement of species from the ground and satellites or in general atmospheric chemistry and physics. Many recent publications have been presenting approaches on the quantification of GHG emissions which we want to briefly summarise here to provide a framing for our own work. We start with briefly highlighting two methods, before providing more details on potential data sets that have been used and could be used to further explore emission strength.

van Damme et al. (2018) focused on industrial and agricultural ammonia hotspots and the calculation of emission fluxes from the distributions of $NH_3$ with a box model, assuming the constant first order loss term E = M/$\tau$, where the emission flux E is calculated directly from the total mass M of the species and its atmospheric lifetime $\tau$. Especially with regard to the hotspots, they used a background column, which was subtracted from the $NH_3$ column, to extract the point sources. For more details, see Section 3.2. Dumont Le Brazidec et al. (2022) uses a deep learning image recognition method to approach a similar question. They analyse simulated carbon dioxide fields for a future application on $XCO_2$ measurement data from the TROPOspheric Monitoring Instrument (TROPOMI) on the Sentinel-5P satellite, to assess $CO_2$ emission plumes. An objective threshold choice for processing satellite images is difficult due to the low signal-to-noise ratio. However, they address this problem by a loss-function and a suitable addition of background and instrument noise.

Global coverage with a spatiotemporal resolution good enough to detect localised emission hotspots is a key element for estimating and validating future global inventories. Thus, large datasets that are continuously updated require perpetual analysis with flexible workflows to monitor emissions. Currently, most analyses are very much case specific and use specific data sets. For example, Goudar et al. (2022) explores carbon monoxide from a multi-annual data record of TROPOMI in a well-targeted proof of concept study. In contrast, the study by van Damme et al. (2018) uses an averaged ammonia data set from the Infrared Atmospheric Sounding Interferometer (IASI) and Tu et al. (2021) uses a single year of cloud-free methane total column data measured by TROPOMI as a case study. Of course, data availability is an important limiting factor. Another are transferable methods that can be easily applied to different datasets and are scalable. Both ingredients are required to support reliable future emission estimates of any gas.

A recent approach for the determination of methane emissions from high-resolution satellite observations is by Varon et al. (2022), also using data from the TROPOMI instrument. In the case of van Damme et al. (2018) a subsampling of the data was obtained by regarding the cloud coverage. Because of high gradients between the sources and the background a further subsampling was not necessary. These are special conditions that are not suitable for any show case, especially not on the Earth's surface, due to transport and mixing processes. Many problems arise with transport and mixing processes of relatively long living greenhouse gases like methane and carbon dioxide or high background emissions, which makes it hard to find and quantify emission hotspots.





We define an emission hotspot as a frequently emitting source whose strength (measured as a massflux per unit area and time) is higher than the background emissions. The idea is that such an emission causes a detectable local enhancement of mixing ratios (or related measures) in the lower atmosphere. This enhancement can be detected and monitored over time to estimate an emission strength. The "Routine for the Isolation of Chemical Hotspots in Atmospheric Research Data" (RICHARD) algorithm is a comprehensive approach to identify, highlight, localise and quantify those emission hotspots within big datasets of mixing

ratios (or related measures) of any trace gas.

Within Section 2 we present the RICHARD algorithm and its basic mathematical construction. In principal, RICHARD "slides" through any given spatiotemporal dataset and detects local enhancements using a self-determined threshold value. The threshold value is determined by an iterative procedure that checks if the current value needs to be adjusted or not. This threshold value is called the structure quotient and is defined by two key variables which are determined by all data points in-

and outside any possible hotspot area. For testing purposes of the RICHARD algorithm, we use the atmospheric and chemistry model ICON-ART (Zängl et al., 2015; Rieger et al., 2015; Weimer et al., 2017; Schröter et al., 2018). Here, we implement an arbitrary tracer with predefined emission hotspots and will test RICHARD's ability to identify and quantify the emission hotspots in free running model simulations (e.g. variable meteorology and transport), as described in Section 2.4.

In Section 3 we perform sensitivity tests to evaluate the quality of RICHARD's detection ability. Taking wind conditions into

account for calculations of emission plumes is very effective as shown by Varon et al. (2018) and Heimerl et al. (2022). Here, we show that our algorithm chooses correctly situations when transport plays a minor role - e.g. calm conditions or simple laminar flow. Within this evaluation, wind speed and direction play a key-role for an identification and calculation of the source strength of a hotspot.

Finally, we address the selection of the best possible setup of RICHARD for the quantification of emission hotspots in a

user specific dataset (see Section 3.3). For this purpose we developed a quality function into which one puts the two most important variables in the setup selection, namely the size of the pattern and the selected threshold with which RICHARD selects or rejects the time steps. Depending on the combination of these variables, you end up more or less close to the real source strength, which we know for the given test case of model data.



## 2 Methods

### 2.1 The RICHARD algorithm

RICHARD is a python based program that takes data in netCDF format which are on a regular latitude and longitude grid. This can be global or regional map data, such as climate model output, satellite and ground based measurements. RICHARD is flexible in the horizontal and vertical resolution of the mapped data, but it is recommended to use the surface level (or the level nearest to the actual emission). In this section we will explain the mathematical foundation of the algorithm and the construction of the workflow by going through the components step by step.

First, RICHARD opens the input dataset and selects the specified variable (e.g. methane volume mixing ratios) and a rectangular area given via the latitude and longitude values of its vertices (e.g. our region of interest to search for emissions). This area, hereafter named P0, is partitioned in two so called patterns, named P1 and P2. Pattern P1 contains all the cells that we assume to be part of our expected hotspot area. P2 is the complement so that P1 + P2 = P0. Let $n_1 = \#P1$ and $n_2 = \#P2$ be the number of cells that belong to P1 and P2, respectively. The predefined patterns are now fix for the whole loop, only the input data changes from one timestep to another, e.g. the volume mixing ratio of a chemical tracer. These data values at pixels i in P1 and at pixels j in P2 are named v(i) and v(j) with $i \in \{1, ..., n_1\}$ and $j \in \{1, ..., n_2\}$. We then define the arithmetic mean value m of an iteration step in P0 as:

$$m \quad = \quad \frac{\sum_{k=1}^{n_1+n_2} v(k)}{n_1 + n_2}$$

Furthermore we are now interested in the number of pixels in P1 and P2 that are above the mean value m of the whole rectangle P0. Therefore an indicator is defined which is equal to 1 if the condition v(i) > m is true and 0 if not. We summarize all indicator values in P1 (or P2) to get the number of occurrences above the mean as shown in Equation 1 for P1 and in Equation 2 for P2, accordingly.

$$a_1 \quad = \quad \sum_{i=1}^{n_1} \mathbf{1}\{v(i) > m\} \quad = \quad \sum_{i=1}^{n_1} \mathbf{1}\{v(i) > \frac{\sum_{k=1}^{n_1+n_2} v(k)}{n_1 + n_2}\} \tag{1}$$

$$a_2 \quad = \quad \sum_{j=1}^{n_2} \mathbf{1}\{v(j) > m\} \quad = \quad \sum_{j=1}^{n_2} \mathbf{1}\{v(j) > \frac{\sum_{k=1}^{n_1+n_2} v(k)}{n_1 + n_2}\} \tag{2}$$

With these preparatory calculations done we can define a quotient that sits at the core of the actual algorithm. The so called structure quotient sq contains the relative number of occurrences in P1 that are above the mean value m in the numerator and the relative number of occurrences in P2 that are above m in the denominator as shown in Equation 4.

$$sq \quad = \quad \frac{\frac{a_1}{n_1}}{\frac{a_2}{n_2}} \quad = \quad \frac{a_1 \cdot n_2}{a_2 \cdot n_1} \quad = \quad \frac{\sum_{i=1}^{n_1} \mathbf{1}\{v(i) > \frac{\sum_{k=1}^{n_1+n_2} v(k)}{n_1+n_2}\} \cdot n_2}{\sum_{j=1}^{n_2} \mathbf{1}\{v(j) > \frac{\sum_{k=1}^{n_1+n_2} v(k)}{n_1+n_2}\} \cdot n_1} \tag{3}$$





with

$$i \in \{1, ..., n_1\}, \qquad j \in \{1, ..., n_2\}, \qquad k \in \{1, ..., n_1 + n_2\}$$

For each iteration in reading the dataset, RICHARD calculates the value of sq and in principal there are three outcomes that are possible. First, the structure quotient sq is near to 1 if numerator and denominator are almost equal. If there are relatively
more pixels above the mean in P2 than in P1 (denominator greater than numerator), sq decreases to a value below 1. Just in case there are relatively more pixels above m in P1 than in P2, sq increases to values above 1, which is the desired outcome. As we are about to select iteration steps where values in P1 are significantly higher than outside, the first two alternatives are not interesting for us. Dependent on the threshold value $sq_t \in \mathbb{R}$ RICHARD only keeps iteration steps with $sq > sq_t$ and drops the ones with a structure quotient smaller or equal to $sq_t$. The result is a subset of the original dataset, e.g. 5 % of it, that contains
only time steps of the simulation that fulfill the above mentioned conditions. In particular the region of pattern P1, which we assume to be a hotspot area, is highlighted. More characteristics and properties of subsets that are selected by RICHARD are described in Section 3.

## 2.2 Setup definition

The goal of RICHARD is to highlight hotspot areas and, as explained in Section 2.1, the chosen area is processed inside the
algorithm as pattern P1. Within the definition of this pattern, the RICHARD source code comes with two advantages that are implemented. First, RICHARD is able to recognize the grid resolution of the given input data and the patterns are created according to the mesh size of the individual grid. At the moment this is possible for regular grids only. Second, it is only necessary to provide a text file with comma separated values for longitude and latitude, e.g. locations of factories or offshore platforms, then the program creates the patterns itself on basis of these data. If a point of latitude and longitude falls into a grid
cell, this pixel will be part of P1. Since the difference between emissions over land and sea is usually characterised by a high gradient, RICHARD offers the possibility to switch off one of the two options via a land-sea mask. This is especially useful when a study area contains parts over land as well as over water, as this inhomogeneity of emissions could lead to a distortion of the results. Per default all land surfaces are not taken into account, but users can decide if they want to have the land surface masked or not. This is implemented by the python module global-land-mask (Todd, 2020) which checks whether a point given
by its latitude and longitude values is on land or sea.

Although RICHARD is written in python, users do not need to change code if they want to apply the algorithm to their own model simulation output. The definition of a setup is simply made through a user-friendly command line interface, which was implemented by using argparse, a python module that parses arguments written in the command line and links them with variables used in the source code. The argparse module also creates automatically generated help and error messages, e.g.
when users give invalid arguments. Within this command line interface it is necessary to specify the following directories and information for a RICHARD setup: First, it is mandatory to indicate the complete input directory. This must be a folder only consisting of all single netCDF files that should be processed by RICHARD. From all files in this input directory, the algorithm will select a certain percentage, and copies them into the given output directory. The names of the files can be changed with the





-f flag. Then, it is also mandatory to provide the path to the above mentioned csv file with the information about the location
of the pattern. With the flags −−llat, −−ulat, −−llon and −−ulon one can define the latitude and longitude boundaries for P0. It is
mandatory that P1 is a real subset of this area. Some additional selection can be made concerning the vertical level and the
timestep, both are only necessary if multiple layers or timesteps are within one file. Table 1 shows an overview of the command
line arguments of RICHARD together with their description.

**Table 1.** Overview of command line arguments of RICHARD. Displayed is the short and long version of the argument, the description that
also arises in the help message and the information if this argument is required for RICHARD or not.

| argument (short) | argument (long) | description | required |
|:---:|:---:|:---:|:---:|
| −i | −−indir | input directory | yes |
| −o | −−outdir | output directory | yes |
| −p | −−path_to_pattern_file | path to csv file with lat / lon information for pattern | yes |
| −f | −−filename | name of output files | no |
|  | −−llat | lower latitude boundary | yes |
|  | −−ulat | upper latitude boundary | yes |
|  | −−llon | lower longitude boundary | yes |
|  | −−ulon | upper longitude boundary | yes |
| −l | −−lev | vertical level selection | no |
| −t | −−timestep | iteration step in input files | no |
| −s | −−sq | threshold for structure quotient | yes |
| −m | −−land_mask | use land mask (True or False) | no |

RICHARD comes with a separate tool which is also usable via the command line. The so called threshold analysis aims
to evaluate the input data in advance and provides the threshold values $sq_t$ which is needed to create a subset that contains
a certain percentage of the original dataset. The threshold analysis tool processes the structure quotient for each time step of
the input dataset and writes a text file with the arithmetic mean, median, upper quartile, 0.9-quantile and 0.95-quantile of all
quotients. On the basis of this list, the user knows which value of $sq_t$ to chose for an elimination of e.g. 95% of the original
set. With simple adjustments the user can also integrate own empirical p-quantiles that are written into the output text file. As
listed in Table 1, this threshold value is a mandatory specification to start RICHARD via the command line.

## 2.3 Comparability of setups

For a visualisation of the structure quotient, we want to formulate it as a mathematical function. Therefore we denote the
relative number of pixels above the mean in P1 (formerly $a_1/n_1$) as x and the number of pixels above the mean in P2 (formerly



$a_2/n_2$) as y. If P1 and P2 are well balanced and n → ∞, we can say that x + y = 1 is valid and the structure quotient can be

formulated as

$$y \quad = \quad \frac{x}{1-x} \quad := \quad f(x). \tag{4}$$

Figure 4 shows the graph of this function. We can see, that with a rising percentage of pixels above the mean in P1, the

structure quotient is increasing. If there are no pixels above the mean in P2, this would lead to a special case, where x = 1 and

the structure quotient is infinity. Those iteration steps are, of course, selected by the algorithm, but to avoid bias, we use the

median for mean calculations, which is robust to such outliers.

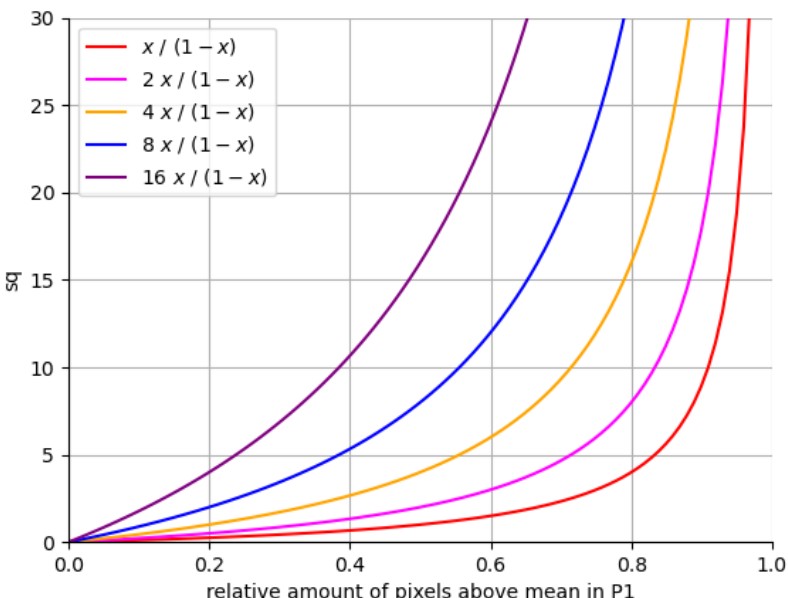

**Figure 1.** The graph of the structure quotient (sq) with different coefficients 1, 2, 4, 8 and 16.

Here, we displayed five different cases of the formulation of the structure quotient and we recognize a squeezing and

stretching of the graph, dependent on the coefficients 1, 2, 4, 8 and 16. With the definition of P1 and P2, the coefficient of

x in Equation 4 varies, what leads to the changes in the graph on the y-axis, where the height of the structure quotient is

denoted. This shows, that the height of sq is relative and dependent to the chosen pattern size and the strength of a hotspot. In

other words: The higher the Delta between a hotspot an the background, the higher the coefficient in Equation 4. With higher

coefficients, we will get higher threshold values and structure quotients, which are only valid for a comparison within a fixed





setup. Hence, structure quotients can only be compared within one graph, not between two or more graphs. This leads to the fact, that if we change the size of P1 and P2, the results concerning the structure quotient might be completely different and,

most important, they are not comparable e.g. by an increase of sq, due to this relative dependency. Later on in Section 3.3 we will take a look at how to choose a optimal setup for a user specific dataset by taking the variables sq and #P1 that we introduced here, into account.

## 2.4 The modeling framework ICON-ART

The atmosphere and chemistry modeling framework ICON-ART (ICOsahedral Nonhydrostatic - Aerosols and Reactive Trace

Gases; Zängl et al. (2015); Rieger et al. (2015); Weimer et al. (2017); Schröter et al. (2018)) is a joint development of the German Weather Service and Max-Planck-Institute for Meteorology in Hamburg. The main goals and characteristics that make ICON an outstanding next generation numerical weather prediction (NWP) and climate modeling system are the exact local mass conservation and mass-consistent transport formulated as non-hydrostatic equations, the scalability on parallel high-performance computing (HPC) architectures and the availability of vertical nested grid such as one-way and two-way

horizontal nesting for spatial resolutions down to 100 m (Zängl et al., 2015; Prill et al., 2019). The calculations of the ICON model are performed on an icosahedral-triangular C grid, which allows to overcome problems with boundary conditions and singularities, e.g. at the poles (e.g. Staniforth and Thuburn, 2012). In addition to that, global and local grid refinement is easily possible within this type of horizontal grid as shown in Sadourny et al. (1968). Vertically, the ICON model reaches a height of 75 km usually subdivided in 90 levels, starting with a lowest model level of 20 m above the ground.

The ART modules developed at KIT extend the numerical weather and climate prediction system ICON with chemistry, aerosol dynamics and radiation feedback processes. The model aims at simulating interactions between the trace substances and the state of the atmosphere by coupling the spatiotemporal evolution of tracers with atmospheric processes (Schröter et al., 2018). ICON-ART is a state of the art chemistry and climate model and represents a perfect environment for the application of RICHARD 1.0.

For a simulation and emission of these hereafter called tracer in global atmospheric chemistry models, many different approaches have been taken in the last decades (e.g. McKeen et al. (1991) or Keller et al. (2014)). ICON-ART reads the emissions which are necessarily given in $kg\,m^2\,s^{-1}$ for areal or point sources from files in the netCDF format. The emissions are converted to mass mixing ratio (MMR) and added to the actual MMR of the corresponding tracer at the lowest model level. While the MMR of the tracer is used for internal computations, the output is given in volume mixing ratio (VMR) in mol

$mol^{-1}$. The detailed workflow of the emissions module in ICON-ART is explained in Weimer et al. (2017).

For this work we chose a global simulation for one complete year with a horizontal resolution of about 160km between the center points of the triangles. The output is written on a regular latitude / longitude grid with $1° \times 1°$ resolution and a temporal resolution of 12 hours. The study area and pattern P0 for the RICHARD algorithm are defined as a square at $0°$-$10°$ E and $52°$-$62°$ N. This section of the earth is located in the middle of the North Sea, which turned out to be a well balanced area

with constantly changing wind directions. This ensures that the algorithm is not disturbed due to a too strong influence of a



prevailing wind direction and wind strength and the results are not biased. Pattern P1, as defined in Section 2.1 is a four times four square in the center of this region.

For the studies and the showcase presented in this work we need an implementation of emissions which is highly spatiotemporal flexible. This is why we decided to use pointsources which can be implemented with various source strengths, alternating heights and different start and end times. For this setup we defined a dummy tracer with a constant source strength of $1 \text{ kg s}^{-1}$ at 10 meters height located exactly in the center of the P0 square.





## 3 Results

### 3.1 Identification of hotspots

The above described ICON-ART simulation consists of 730 time steps for one year, which will be processed within RICHARD.
By taking a closer look at the single time steps that were chosen or rejected by the algorithm, we confirmed a strong dependency
of the identified concentration field and the wind conditions. In Figure 2 we can see that for windless conditions, the emission
hotspot is clearly visible (a), whereas the substance is transported away from the original emission source by high wind speed
into the lower right corner (b). Here, the pattern P1 was chosen as a four times four square around the original emission source,
which is marked with a magenta colored cross. The time step shown in Figure 2 (a) is selected by the RICHARD algorithm
because of its high structure quotient (relatively high values inside of P1), while the time step in (b) was rejected, due to high
values outside the pattern.

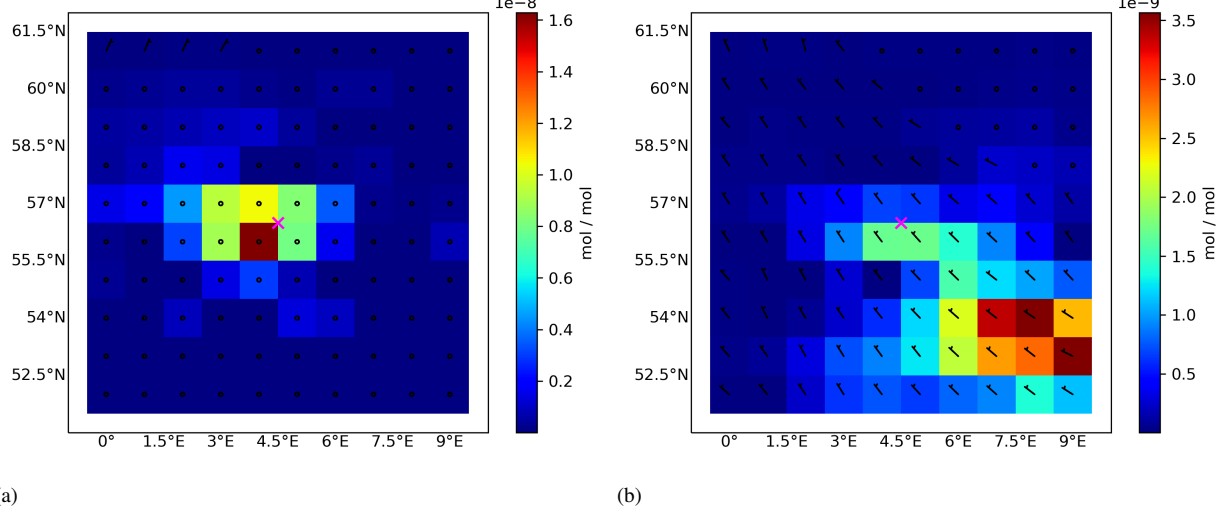

(a)                                                          (b)

**Figure 2.** Single time steps from the ICON-ART simulation showing the volume mixing ratio of the dummy tracer for windless situations
(a) and conditions with relatively high wind speeds (b). The emission source is marked by a magenta colored cross.

The wind barbs, as displayed in Figure 2 represent wind direction and speed. The direction from which the wind is blowing
is indicated by the direction of the shaft of the wind barb points. For example, if the shaft points to the right, it indicates the
wind is blowing from the west to the east. The wind speed is indicated by the number of feathers or barbs attached to the shaft,
where one barb represents a wind speed of 5 knots. Circles indicate meteorological situations without wind speed or direction.

From the mathematical construction of the algorithm, the pixels that are in P1 have to be defined for each run of RICHARD,
which would result in a brute-force method for an unknown emission source pattern. Thus, we now address the question how
we can enable the algorithm to determine P1 pixels and to find emission hotspots on its own. In other words, until now we
did not discuss how the pixels of P1 and P2 will be chosen by RICHARD, which, of course, is affected by the concrete setup





and emission source that one wants to measure. For this purpose, we set up 8 test cases in which a block of 4x4 pixels (P1) is positioned differently within a 10x10 pixels block (P0). Here, P1 is placed in the upper, middle and lower left, center and right of the 10X10 block. The emission source stays in the exact center of P0, only the location of P1 changes as described above. We want to test, if we can force RICHARD to find high concentrations outside the center box and so the information about where the emissions come from, is not directly provided to the algorithm. RICHARD selected the all time steps of the original

simulation with a structure quotient higher than the 95%-quantile. The mean of all selected time steps and the location of P1 in the 8 test cases (magenta crosses) are displayed in Figure 3.

The results of the test cases show, that RICHARD is able to identify concentrations within each of the directions. We were able to force the algorithm, by a targeted choice of patterns, to find and highlight concentrations of the dummy tracer which are about 10 times higher as in the rest of the square, even if the actual source is located in the center of the test area and not in

the corresponding P1. Nevertheless, two things are remarkable in this context: First, the original emission source in the middle shows the highest concentrations in all test cases. If just sensitive enough, we could find the original source without knowing the concrete location, just by selecting patterns near to it. Second, we recognize, that for each of the 8 test cases, the wind direction is pointing to the corners in which the corresponding P1 is located. This matches to our findings with the single time steps from the beginning of this Section and displayed in Figure 2. Only in the case that the locations of the pattern P1 and the

actual emissions source are identical, we observe windless conditions in our model. This shows a clear influence of the wind speed and direction on the selection made by RICHARD.

Patterns of high concentrations may be found by RICHARD, which is a nice proof of concept for the algorithm itself, but only for real hotspots - which we defined as a frequently emitting source with a relatively high delta to the background - we get a clear image of the emission plume, as visible in Figure 3 (e).

**Table 2.** Summary of the eight test cases. Displayed are the value of sq, which was taken as a threshold for the respective RICHARD selections such as the structure quotient of the original and the selection results.

| test case | threshold value | sq of original result | sq of RICHARD selection |
|---|---|---|---|
| upper left | 1.97 | 0.48 | 4.63 |
| upper center | 1.24 | 0.23 | 2.47 |
| upper right | 0.23 | 0 | 0.79 |
| left center | 4.68 | 4.44 | 7.88 |
| center | 8.63 | 6.20 | 14.70 |
| right center | 1.67 | 1.05 | 3.75 |
| lower left | 1.61 | 0.75 | 3.09 |
| lower center | 1.50 | 1.75 | 2.63 |
| lower right | 0.72 | 0.48 | 2.49 |





**Figure 3.** Displayed the means of the selections made by the RICHARD algorithm for the corresponding 95 % quantile threshold (see Table 2) together with the wind direction and speed (black barbs) such as the locations of the different P1 patterns (magenta crosses) in the 8 test cases.

Table 2 summarizes the threshold values that were used in RICHARD to select 5 % of the data together for the 8 test cases. Additionally the structure quotients that arise for the original mean and the selection mean, using the corresponding pattern P1 (magenta crosses in Figure 3) are displayed. For the thresholds, we find most of the values between 1 and 2, which is far below the threshold of the original P1, which is 8.63. This difference is a first indication about the location of an emission





source, because with a frequently high gradient between P1 and P2, it is more likely to have high structure quotients as well.
Two remarkable outliers arise at the upper right corner, with a weak sq of 0.23 and in the left center, with a strong 4.68. For both, also the structure quotient of the original and the selection mean state the lowest / highest result. We performed the same tests with the thresholds that arise from the upper quartile of the original data (instead of the 95 % quantile) and found similar results, with a little weaker signals for the 8 cases, probably due to the increasing of the number of time steps.

In this section we showed, that the RICHARD algorithm is capable for identifying areas with local enhancements and
especially emission hotspots. We found out, that wind conditions play an important role in the identification, as only for hotspots of actual emissions we recognized zero wind, whereas the found patterns with relatively high tracer concentrations are caused by randomly fitting wind events.

## 3.2 Structure quotients and source strengths

Within this section we discuss the selection procedure of the optimal threshold value for a custom dataset. In general, for the
threshold value, we select a percentage of the time steps with the highest structure quotients within a given dataset. We could, of course, select only a handful of iteration results with the best sq, but this would lead to a very limited selection and mean calculations or time information would be completely biased. We therefore have to take a closer look on the distribution of structure quotients as a function of the threshold values. Selecting all iteration steps would bring us the mean structure quotient, so we obviously find the optimal threshold value below this boundary. For the choice of the best threshold value, a compromise
between a high structure quotient and a sufficiently large number of iterations that we want to have available is necessary, and as mentioned above, the exact quantification of the source strength is the most important factor.

By definition, we know the exact source strength that is emitted, which is exactly $1 \, \mathrm{kg \, s^{-1}}$. So for the next step, we can trace back from the volume mixing ratio in $\mathrm{mol \, mol^{-1}}$ to the (known) emission flux. We can therefore determine, which threshold value for the structure quotient is optimal for a source quantification in our test case. In a first step, from the VMR we derive the
MMR, as it is used for internal calculations in ICON-ART. We therefore use the molar weight $M$ of our artificial tracer, which is $0.03 \, \mathrm{kg \, mol^{-1}}$, and of dry air, which is $amd = 28.970 \, \mathrm{g \, mol^{-1}}$. Equation 5 shows the conversion from VMR to MMR.

$$\mathrm{MMR} \quad = \quad \frac{\mathrm{VMR} \cdot \mathrm{M} \cdot 1000}{\mathrm{amd}} \quad \left[ \frac{kg}{kg} \right] \tag{5}$$

Next, we calculate the mass of the air column in $\mathrm{kg \, m^{-2}}$ in every column. As the ICON-ART model consists of 90 vertical levels, we take

$$\mathrm{kgair(i)} \quad = \quad \frac{\mathrm{p(i) \cdot h(i)}}{\mathrm{R \cdot T(i)}} \quad \left[ \frac{kg}{m^2} \right] \tag{6}$$

as the total column air mass. Herein, $R = 8.314409 \, \mathrm{J \, K^{-1} \, mol^{-1}}$ denotes the ideal gas constant, p(i) the air pressure in Pa and T(i) the air temperature in K in the i-th height level h(i), with i $\in$ [0,89]. Although the original approach of Weimer et al. (2017) assumes emissions to the lowermost level only, we take the complete vertical column for two reasons: First,





we want to derive emissions using local VMR enhancements from many subsequent timesteps, during which the tracer has
been transported under certain conditions, both horizontally and vertically. If we would omit those higher levels, the results
(regarding the column enhancement) might be distorted. Second, this is also much more senseful, when applying RICHARD to
satellite observations, where a majority of the measured gases are found near the ground, but another big amount in the higher
regions would be lost.

We now use the derived MMR and the total column air mass from Equations 5 and 6 such as the time between two output
timesteps $\Delta t = 12\,\text{h} = 43200\,\text{s}$ to calculate the emission flux $E$ as

$$E \quad = \quad \frac{\text{MMR} \cdot \sum_{i=0}^{89} \text{kgair(i)}}{\Delta t} \qquad \left[ \frac{kg}{s} \right] \tag{7}$$

The resulting emission source strength E in $\text{kg s}^{-1}$ is expected to be near to the implemented flux of $1\,\text{kg s}^{-1}$, as we applied
the formulae of the original ICON-ART code for the calculations. For the mean of the full simulation, we get an emission flux
of $0.625\,\text{kg s}^{-1}$ for the source at $5°\,\text{E} / 56°\,\text{N}$. For the mean of the simulation performed in Section 3 the structure quotient
is 6.2, which is already a relatively high value and can be explained by the zero background emissions in this setup. We now
choose five different threshold values for our model dataset to find out the most suitable subset to derive the exact emission
flux. These are the median, the mean, the upper quartile, the 95 % quantile and the 99 % quantile of all 730 threshold values of
the simulation and let RICHARD select with these conditions. We find the upper quartile as the best fitting subset, where the
structure quotient of the result is 14.7 and therefore factor two better than in the original dataset without a RICHARD selection.
Also the quantification of the emission source strength shows the best results within the selection of RICHARD using the upper
quartile threshold. We derive a source strength E of $0.983\,\text{kg s}^{-1}$ which is an underestimation of less than 2 %, and, compared
to the original mean of $0.625\,\text{kg s}^{-1}$ a much more accurate result. An overview of the five test cases and the mean of the full
simulation can be found in Table 3.

**Table 3.** Summary of the six test cases. Displayed are the value of sq, which was taken as a threshold for the respective RICHARD selections,
the structure quotient of the results and the estimated source strength.

| test case | threshold value | sq of RICHARD selection | estimated source strength [$\text{kg s}^{-1}$] |
|---|---|---|---|
| full | - | 6.2 | 0.670 |
| median | 3.61 | 11.38 | 0.833 |
| mean | 4.39 | 11.38 | 0.973 |
| upper quartile | 4.85 | 14.7 | 1.055 |
| 95 % quantile | 8.63 | 14.7 | 1.858 |
| 99 % quantile | 12.6 | 26.25 | 1.812 |

The increase of the resulting structure quotient from 6.2 to 14.7 marks an enhancement of about 137 %, as for the source
quantification, we were able to reduce the error from 37.5 % to 1.7 %. Although the 99 % quantile shows a bigger increase in





the resulting structure quotient, the source strength is largely overestimated, as the selection in this case is probably too small and therefore not meaningful.

Here, we used the above explained artificial tracer within the simulation of the ICON-ART model. The calculation of the source strengths from the VMR enhancement of the substance, as shown in this Section, depends on some particular properties, which we need to address within an application tailored for a certain gas (like, for example, methane). First, the artificial tracer in this work has a zero background. Thus, no subtraction of a background value is necessary (but can be easily included). Second, we looked at an isolated source. Thus, the signature of other nearby emission sources did not interfere. Third, no chemical reactions with other substances were simulated for our artificial tracer. Thus, we looked at an infinite lifetime, which for an annual integration for a long-lived tracer is reasonable - otherwise chemical sinks and sources will affect the estimate of emission source strength. For an application of the method on more complex model simulations with GHG or satellite observations, test cases in areas with low background values and an adjustment of Equations 5, 6 and 7 are necessary.

### 3.3 Measuring the quality of a setup

The findings of the previous Sections inevitably lead to the question about the quality of a chosen setup and, of course, how to find the optimal setup for a user-specific dataset. This depends on two main factors. First, as we already evaluated in the previous Section, the choice of the p-quantile for a threshold of sq within RICHARD. Second, the variation of the size of P1. So far, we did not change the pattern size, and always used 16 pixels. With the choice of the p-quantile and the variation of the size of P1, we now determine, how the estimated source strength depends on the chosen parameters and what the optimal choice of parameters is. We therefore define g(p, $n_1$), the quality-function of RICHARD, as the estimated source strength, depending on the size of P1 and the chosen p-quantile as a threshold for the structure quotient. We put the p-quantiles from 0.5 to 0.99 on the x-axis, the size $n_1$ of pattern P1 from 2 to 80 on the y-axes and the difference of the RICHARD-estimated and the original source strength on the z-axis of a graph. As displayed in Figure 4 we get a 3D surface, showing the connection of the three variables.



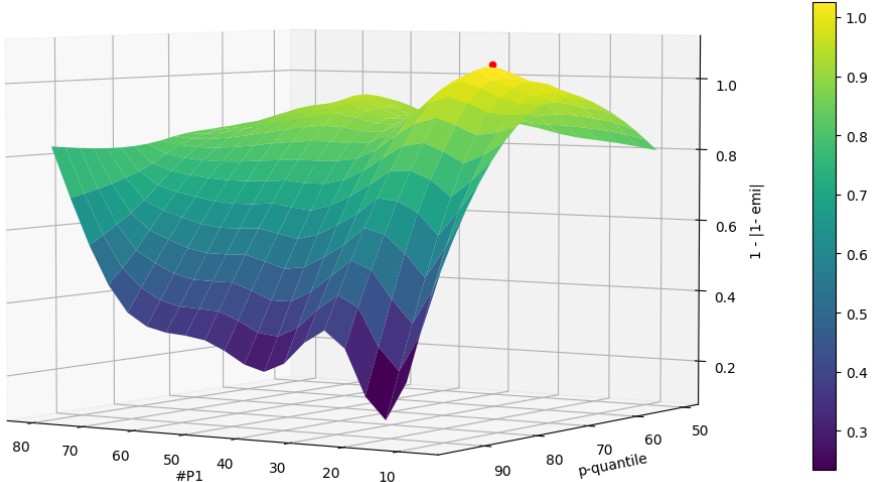

**Figure 4.** The 3D surface shows the graph of g(p, $n_1$), the connection of the selected p-quantiles from 0.5 to 0.99 on the x-axis, the size of pattern P1 from 2 to 80 on the y-axes and the difference of the RICHARD-estimated and the original source strength on the z-axis. The red dot marks the setup, which was chosen for this work, namely #P1=16 and the 0.75-quantile.

We recognize good estimates for the correct emission source strength of $1.0\ \mathrm{kg\ s^{-1}}$ between the 60 and 80 %-quantile and
for #P1<30. For p-quantiles with p > 0.8 and almost all chosen sizes of P1, the estimates are getting worse, so we find the optimal setup for our dataset at g(0.75,16), hence for the upper quartile as a threshold for the subset and #P1=16 as the size of pattern P1, marked with a red dot in Figure 4. With an estimation of $1.055\ \mathrm{kg\ s^{-1}}$, this setup has a deviation of only 5.5 % to the input strength of $1\ \mathrm{kg\ s^{-1}}$.





## 4 Conclusions and Outlook

RICHARD 1.0 is a comprehensive tool for the identification and quantification of emission hotspots and uses a novel methodology and workflow, including spatiotemporal proxy data and a selection algorithm. It comes with an implemented command line interface for a user friendly application. Also the pattern definition is automated with low hurdles for a quick start to use the method with all kind of (dense) geo-referenced data. We defined a structure quotient to analyse the selected area in terms of the mixing ratio of the tracer in the previously defined pattern, relative to the ratios found outside the chosen subset. In general,

the algorithm decides on behalf of a threshold value for sq, if a time step of the dataset meets the criteria of being a hotspot and if it is kept for further calculations or not. An analysis function that is implemented evaluates the data in advance and suggests user- and dataset-specific threshold values for this routine.

We used simulation output of the ICON-ART model as test data to show, that the structure quotient, as mathematically constructed in the RICHARD algorithm, is able to detect windless situations. Only with matching pattern and hotspot point

sources, we recognized zero wind, whereas identified areas with relatively high tracer concentrations were caused by randomly fitting wind events. For our ICON-ART model dataset, we found the upper quartile as the best compromise between a high sq of the results, sufficiently large selection-datasets and the precision of the calculated emission fluxes. To get a clear idea of how to measure the quality of a chosen setup, we defined the quality function $g(p, n_1)$, that projects the estimated source strength, depending on the size of P1 and the chosen p-quantile as a threshold for the structure quotient. For our optimal setup, we were

able to quantify the emission hotspot in the model by RICHARD with an error of about 5 % to the given input source strength, which is about 28 % better than without the algorithm.

We will soon explore, if the ability of RICHARD to identify unknown emission sources can be helpful within inverse modelling and data assimilation purposes. The importance of new techniques within the development of inverse models is highlighted in many publications in the past years (e.g. Henne et al. (2016), Yu et al. (2021)).

This work serves as a proof of concept for a later application of the method on satellite measurement data, as shown e.g. in van Damme et al. (2018). Here, as explained in Section 3.2, we need to add a viable tracer independent way to subtract background values, to obtain information about point sources that cause local enhancements (to a non-zero background). As a link to further experiments with measured atmospheric data, an application of the algorithm to model data with emission fluxes at a show case region of the greenhouse gas emission inventory EDGAR (Emission Database for Global Atmospheric Research,

Janssens-Maenhout et al. (2019)) is planned to show that this method is successfully working on non-synthetic data and on a bigger scale as well. Furthermore we will implement RICHARD 1.0 into the WALLACE (Workflow for the Adjustment of Low Level Atmospheric Compounds and Emissions) workflow (Scharun, 2022) which aims to be a multistage tool to quantify and model emission hotspots and evaluate their impact on the regional and global GHG budget.



*Code availability.* The recent version of the code of RICHARD and the RICHARD threshold analysis was published by Scharun (2023a)
and is available via KITopen: https://doi.org/10.5445/IR/1000158327

The ICON-ART model simulation and csv file as an input dataset for RICHARD and the RICHARD threshold analysis were published
by Scharun (2023b) and are available via KITopen: https://doi.org/10.5445/IR/1000159472

.

*Author contributions.* Christian Scharun, Roland Ruhnke and Peter Braesicke developed the research question. Christian Scharun wrote the
manuscript and performed the data analysis with input from Roland Ruhnke and Peter Braesicke.

*Competing interests.* The authors declare that they have no conflict of interest.

*Acknowledgements.* Acknowledgements go to the Helmholtz project Digital Earth for funding this work.

This work was performed on the computational resource HoreKa funded by the Ministry of Science, Research and the Arts Baden-
Württemberg and DFG (Deutsche Forschungsgemeinschaft).



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
