# Peer review of "RICHARD 1.0 - Routine for the Isolation of Chemical Hotspots in Atmospheric Research Data"

_Geoscientific Model Development, 2023_

## Referee Comment (RC1)

The manuscript titled "RICHARD 1.0 - Routine for the Isolation of Chemical Hotspots in Atmospheric Research Data" authored by Christian Scharun et al. introduces a Python module named RICHARD designed to locate, highlight, and measure emission hotspots.
They employ in particular a thresholding technique to isolate areas of high concentrations (due to emissions of hotspots).

The research question the authors aim to tackle—identifying plumes from emission hotspots and estimating their corresponding emissions—is both timely and pertinent.

However, as it stands, the treatment of this question in the article doesn't meet the standards typically expected for a publication in GMD.
Given the range of issues and inadequacies outlined in this review, I find myself compelled to recommend that the paper, in its current iteration, be declined for publication.

Several critical aspects must be addressed, which will be elaborated upon in the following paragraphs. Additionally, after addressing these key concerns, some specific comments (which may or may not relate to the main points) will also be provided.

**First main comment**:
The manuscript, in its current form, is marred by numerous grammatical errors, flawed formulations, and spelling mistakes which hamper its readability and comprehension. Here's an example of a challenging passage:
"In Section 3 we perform sensitivity tests to evaluate the quality of RICHARD's detection ability. Taking wind conditions into account for calculations of emission plumes is very effective as shown by Varon et al. (2018) and Heimerl et al. (2022). Here, we show that our algorithm chooses correctly situations when transport plays a minor role - e.g. calm conditions or simple laminar flow. Within this evaluation, wind speed and direction play a key-role for an identification and calculation of the source strength of a hotspot."

This paragraph is difficult to interpret. The phrase "calculation of emission plumes" is unclear - does it refer to the quantification of emissions related to plumes depicted in the images?

The sentence "our algorithm chooses correctly situations when transport plays a minor role" is also vague and fails to communicate its intended message effectively.

Another confusing point is the stated objective of RICHARD, which is to "identify and quantify emission hotspots." It wasn't until later in the paper that I realized RICHARD's goal was to identify plumes resulting from hotspot emissions. Initially, I was under the impression that RICHARD was pinpointing the hotspots that were producing the plumes. In either case, the phrase "identifying emission hotspots" is misleading and needs clarification.

**Second main comment:**

The method outlined in the paper is challenging to comprehend. It remains unclear how P1 and P2 are initially established. Are they chosen randomly? The statement "Pattern P1 contains all the cells that we assume to be part of our expected hotspot area" seems to indicate otherwise. However, this is an input to RICHARD … So is this an initial guess?

From what I can decipher, P1 and P2 are identified, and then there's an iterative process (which is not explained) to adjust the values of P1 and P2. But when does this process end? What we do know is that the goal is to increase the structure quotient (sq), so the algorithm primarily selects high values from P2 and transfers them to P1, and vice versa for low values in P1. As explained, this appears to be a rudimentary algorithm, and I'm struggling to see what makes RICHARD different from a simple thresholding method.

An other thing: the paper states that RICHARD requires information such as factory or offshore platform locations, and the program creates the patterns based on these inputs. However, what exactly does the term "patterns" refer to in this context?

**Third main comment:**

The authors must justify the value of RICHARD, both theoretically and practically.

From a theoretical standpoint, they should delineate how RICHARD outperforms other methods such as simple thresholding techniques, unsupervised methods like k-means, or supervised techniques like random forests. It is essential to provide a lucid exposition detailing the strengths and unique capabilities of RICHARD over these existing methods.

On a practical level, a comparative analysis against these other techniques, many of which have been implemented in previous studies, is really needed to validate RICHARD's performance.

**Fourth main comment:**

I suggest that the authors apply their method to a real-world dataset, such as TROPOMI data, for example.

**Specific comments**

**section 2.2 Set-up** :
The detailed operational mechanics of the algorithm would be better placed in supplementary material or documented within a notebook or in a Github repository.
Conversely, this manuscript does not provide a clear and concise description of the inputs, outputs, and their respective dimensions, which is a significant oversight.

**Line 159:**
« If P1 and P2 are well balanced and n → ∞, we can say that x + y = 1 is valid »

The term "well balanced" used here is ambiguous. It could be inferred that x+y=1 is applicable

when a1=n1/2 and a2=n2/2, but there is no clear explanation why such a condition would hold true. This point needs to be elucidated.

**Lines 161 – 165:**
The paragraph explains that sq = (a1/n1) / (a2/n2) = x/y, and if y=0, then sq becomes infinite. It is unclear why the authors detail such a basic mathematical consequence.

**Lines 201-203:**
« For this work we chose a global simulation for one complete year with a horizontal resolution of about 160km between the center points of the triangles. The output is written on a regular latitude / longitude grid with 1◦ × 1◦ resolution »

This section refers to 'triangles', which are introduced without context or explanation. The mention of a 160km resolution followed by a 1-degree resolution is confusing. It is not clear if these are equivalent or separate resolutions.

**Line 220:**
« Here, the pattern P1 was chosen as a four times four square around the original emission source, which is marked with a magenta colored cross. The time step shown in Figure 2 (a) is selected by the RICHARD algorithm because of its high structure quotient (relatively high values inside of P1), while the time step in (b) was rejected, due to high values outside the pattern »

This part of the manuscript does not clearly explain the role of RICHARD. Is the algorithm merely accepting or rejecting hotspot locations (represented by the magenta cross) based on the concentration levels nearby or further away? How does this distinguish RICHARD from a basic thresholding algorithm?

**Line 233:**
« We want to test, if we can force RICHARD to find high concentrations outside the center box and so the information about where the emissions come from, is not directly provided to the algorithm »

It seems like the results section should have started with this problem. The experiments performed prior to this point seem to lack context or clear objectives.

**Figure 3:**
This figure doesn't provide clear information about what it aims to demonstrate. The method's initial goal was to 'localise emission hotspots', suggesting the identification of the hotspot source. However, here it seems that the method identifies areas of high concentration resulting from hotspot emissions. It's unclear how this differs from a simple thresholding method that filters for high values.

**Lines 259 – 263:**
This paragraph is particularly difficult to understand and needs to be clarified.

**Lines 290-310:**
The authors test numerous thresholds and derive a range of emission fluxes. They select the threshold that results in an emission flux closest to the 'truth' (<2% relative error), claiming this as

their method's accuracy. This appears to be a 'data leakage' situation, where the method is adjusted to match the known outcome. This is a significant flaw, as the authors should separate their data into a training set to fit their method and a testing set to validate the accuracy of the method. As it stands, the paper doesn't mention any such process.

**Data+code repo**:
The authors would greatly enhance the reproducibility of their results and the usability of their technical tool by providing a notebook file. Considering that the main thrust of this paper is to introduce an easy-to-use tool, a notebook would significantly facilitate this objective.